# Religiosity, Emotions and Health: The Role of Trust/Mistrust in God in People Affected by Cancer

**DOI:** 10.3390/healthcare10061138

**Published:** 2022-06-18

**Authors:** David Almaraz, Jesús Saiz, Florentino Moreno Martín, Iván Sánchez-Iglesias, Antonio J. Molina, Tamara L. Goldsby, David H. Rosmarin

**Affiliations:** 1Department of Social, Work and Differential Psychology, Complutense University of Madrid, 28223 Madrid, Spain; fmoreno@psi.ucm.es (F.M.M.); antmolin@ucm.es (A.J.M.); 2Department of Psychobiology & Behavioral Sciences Methods, Complutense University of Madrid, 28223 Madrid, Spain; i.sanchez@psi.ucm.es; 3Department of Family Medicine and Public Health, University of California San Diego, La Jolla, CA 92093, USA; tgoldsby@health.ucsd.edu; 4McLean Hospital, Harvard Medical School, Belmont, MA 02478, USA; drosmarin@mclean.harvard.edu

**Keywords:** religiosity, emotions, social support, health, cancer patients

## Abstract

Trust in God implies the conviction that God looks after a person’s own interests. The first evidence of a relationship between this construct and people’s psychological and emotional health dates back several centuries. However, the literature on this is limited, especially for people with physical health conditions, such as cancer. Therefore, the purpose of this study is to test the relationships between trust/mistrust in God, social support and emotions in people affected by cancer. The sample consisted of 177 women and men in Spain diagnosed with cancer. The instruments used were the Trust/Mistrust in God Scale, the Positive and Negative Affect Schedule and the Multidimensional Scale of Perceived Social Support. Correlation analysis and hierarchical regression analysis were performed to compare several explanatory models for the dependent variables: positive and negative emotions. The results show significant relationships between all variables. It was observed that, when trust/mistrust in God is included in the model, only mistrust in God predicts both types of emotions. In addition, both social support and some sociodemographic variables help to predict the dependent variables. This study shows that valuing the religiosity and spirituality of oncology patients in healthcare settings can have a significant positive impact on the health of these individuals. Moreover, it represents an important approach to the study of trust/mistrust in God in the context of a traditionally Catholic country.

## 1. Introduction

Estimates indicate that approximately 18.1 million new cases of cancer were diagnosed around the world in 2020, and that this figure will increase over the next two decades to 27 million new cases per year [1]. In Spain, in particular, the situation is equally worrying. According to the Spanish Society of Medical Oncology [1], it is estimated that Spain will reach 280,100 diagnosed cases of cancer in 2022, with a projected incidence of 341,000 cases in 2040. In summary, both in Spain and worldwide, there is an upward trend in cancer cases, which makes this disease one of the main causes of morbidity and mortality in the world.

Cancer, as a physical pathology, requires a medical treatment that depends on its location and stage [2]. However, cancer does not only include medical aspects, but also involves a series of psychosocial aspects that are present from diagnosis to the end of the disease process. For instance, it has been proven that social support plays a very important role for oncology patients [3,4].

### 1.1. Religion, Spirituality and Health: A Brief Contextualization

In an attempt to comprehensively understand people’s health, many authors have been interested in introducing religious and spiritual variables in their studies. In fact, research on religion, spirituality and health has grown rapidly over the past 35 years [5]. In this sense, an upward trend of interest in this field can be observed, as a very significant increase in the number of publications has been found since 1999, particularly after 2009 [6]. This is equally being reflected in the importance researchers are placing on the psychosocial aspects of religion and spirituality (R/S) within medical practice and healthcare settings [7,8,9,10,11], due to their influence on people’s physical and mental health.

Specifically in Spain, interest in the field is not as broad, although there are researchers that have studied R/S in relation to various aspects of mental health [12,13], health behaviors [14] or physical health [15,16] in the Spanish context. The paucity of literature on these relationships in the context of this country may be surprising, given the deep-rooted Christian Catholic culture in Spain, whose values in many cases are closely related to health care. This revolves around the idea that the object of Christian faith is a God of life and, therefore, Christians must assume as their main religious task to promote life in their social and community context, as described by Martín-Baró [17], a Spanish psychologist and priest who is a reference in the study of Social Psychology.

Within healthcare settings, it is important to understand the concepts of religion and spirituality in order to provide holistic patient care [18]. The terms religion and spirituality have been defined in multiple ways. Particularly when it comes to spirituality, there have been deep debates that relate to understanding the concept [19]. In this study, we adopt what Koenig et al. [20] refer to as a traditional view of spirituality, in which spirituality is construed as a characteristic of deeply religious people, which separates them from those who are only superficially religious. Even so, it is important to note that there are conceptions of spirituality that do not necessarily associate it with religiosity. There is a modern view of spirituality, according to which a person does not necessarily have to be religious, but can be a “spiritual, but not religious” person [20]. There would also appear to be what Koenig et al. [20] call a tautological view of spirituality, which, although similar to the previous one, adds to spirituality positive mental health and human values, such as optimism or connection with others. Finally, a modern clinical view of spirituality has emerged, which considers not only religion and positive mental health, but also the secular, so that it considers all people spiritual [20].

In general, “spirituality is a way of being in the world in which a person feels a sense of connectedness to self, others, and/or a higher power or nature; a sense of meaning in life; and transcendence beyond self, everyday living, and suffering” [21] (p. 93). However, religion is understood as an organized system of beliefs, practices, rituals and symbols designed to facilitate closeness to the sacred or transcendent [20]. Moreover, in relation to these terms, religiosity also appears. As stated by Salgado [22], religiosity is the behavioral expression of the system of beliefs, doctrine and organized cults of religion, experienced socially as a body of knowledge, behaviors, rituals, norms and values that govern or are intended to govern the lives of people interested in linking with the divine. In other words, religiosity is the way in which each person expresses their religious beliefs. This term therefore implies, unlike spirituality, a relationship with God.

These concepts include very diverse aspects, which have been studied in relation to health. For example, R/S has been studied with regard to the impact on health of religious support [23], spiritual experiences [24], religious/spiritual coping [25], attachment to God [26], praying [27] or going to church [28].

### 1.2. Trust/Mistrust in God: What Is It and How Does It Relate to Health?

Rosmarin et al. [29] describe trust in God as a construct that involves the conviction that God looks after a person’s self-interest. This conviction would include three fundamental beliefs about the Divine: (a) God is omniscient (i.e., has constant regard for all worldly affairs); (b) God is omnipotent (i.e., is the ultimate power in the universe); and (c) God is omnibenevolent (i.e., is merciful, generous and righteous) [30]. This is to say, since it implies a relationship with God, trust in God is an aspect that is part of religiosity. Conversely, mistrust in God involves the belief that God is not omniscient, omnipotent and omnibenevolent, but is willfully ignorant and malevolent [31].

One of the main ways the relationship between R/S and health is found exists in the power of trust in God as a religious/spiritual coping strategy in the face of stressful situations [29,32,33]. Indeed, trust in God may be an effective coping mechanism in stressful health situations, such as dealing with cancer [34,35], HIV [36] or COVID-19 [37].

On the one hand, following this line, important works in the field have tried to establish relationships between physical health and religious and/or spiritual variables [20]. Krumrei et al. [38] proposed a possible conceptual framework regarding the manner in which trust/mistrust in God might relate to physical (and mental) health through religious coping and found strong positive correlations between trust in God and physical health, as well as negative correlations between mistrust in God and physical health. However, trust in God was not found to be a predictor of physical health, although high levels of intrinsic religiosity increase the magnitude of these relationships [38].

Research on trust/mistrust in God and mental health is, on the other hand, much more extensive. The relationship between trust in God and lower levels of depression, stress and anxiety has been widely documented, as well as the opposite in the case of mistrust in God [30,38,39,40,41].

Particularly, Rosmarin et al. [31] propose that the core beliefs involved in trust in God and mistrust in God may reduce or increase perceived risk appraisals, which impact stress, anxiety and worry. For its part, trust in God may generate positive cognitions about the future, leading to a decrease in hopelessness and depression and an increase in happiness [31]. In short, the work of Rosmarin et al. [31] shows that trust in God is associated with lower levels of depression, anxiety, stress and worry, and higher levels of happiness, while the opposite is true for mistrust in God.

In any case, the assumption that trust in God is a theory-based religious variable that is conceptually linked to emotional and affective states is evident in the aforementioned studies [31].

### 1.3. Emotions and Trust in God: Is There a Relationship between the Two?

The relationships between R/S and emotions have been extensively studied and, generally, findings show positive correlations between religious/spiritual variables and positive emotions, the opposite being true for negative emotions [42,43,44]. Specifically, the relationship between trust in God and human emotions can be traced back to the work “Duties of the Heart” by Rabbi Bachya Ibn Pakuda [45] who, in the context of the Jewish religion, establishes a series of theoretical links between Jewish religiosity (specifically trust in God) and psychological and emotional health. Despite this early relationship, the literature in this area is quite scarce. The main reason for this may be that trust in God has been studied only within Jewish religious contexts [31]. In fact, outside of this religion, research on the subject is practically non-existent.

There is, however, a study examining trust in God concerning Islam. In this religious context, Fadardi and Azadi [46] found results regarding positive relationships between trust in God and positive emotions, as well as negative relationships between the former variable and negative emotions, that support the association between trust in God and mental health indices.

Finally, we cannot overlook the well-known influence that some sociodemographic aspects have on these variables. One of the primary sociodemographic variables that needs to be taken into account is age. Different studies have found that age shows a direct association with different religious/spiritual variables [47,48]. In addition, research has found that other relevant sociodemographic variables, such as socioeconomic status or educational level are inversely associated with religion/spirituality [49]. Regarding emotions, it has been observed that age correlates positively with positive emotions and negatively with negative emotions [50]. Likewise, positive emotions correlate positively with socioeconomic status [51], while educational level is directly associated with positive emotions and inversely with negative emotions [50,52].

In any case, an improved understanding of these relationships is necessary to clarify how trust/mistrust in God influences human health.

### 1.4. Emotions, Trust in God and Health: What about Cancer?

Undoubtedly, in the relationship between trust in God and physical health, positive emotional states can be an important factor, since they have a positive impact on physical health [53]. Among the various existing physical pathologies, researchers have studied trust in God as a coping method in the face of cancer [34,35,54].

In this sense, and focusing attention on the emotional aspect, numerous authors have studied the effects that both positive and negative emotions have on the health of oncology patients. For example, Milbury et al. [55] have found that low levels of positive emotions are strongly associated with lower quality of life in people with cancer. Along these lines, it has been observed that lower levels of positive emotions correlate with greater psychological distress in these individuals [56]. With regard to negative emotions, Wesley et al. [4] have found evidence that this type of affect correlates with higher levels of physical symptoms in oncology patients. Likewise, positive emotions have been found to reduce cancer-related fatigue [57]. In any case, the beneficial effects of positive emotions and the detrimental effects of negative emotions on the physical and mental health of oncology patients has been well documented [58,59,60]. Moreover, it has been observed that there is a relationship between high levels of positive affect and better psychological adjustment of people to cancer [61,62].

### 1.5. The Role of Social Support

Social support has been generally defined as valuing, belonging and tangible support received from one’s own social network [63]. The influence of this support on physical and mental health has been widely recognized [64,65,66]. With regard to the affective aspect, Lakey et al. [67] found that perceived social support is significantly positively associated with positive emotions and negatively associated with negative emotions, with effective predictive power on these variables. Similarly, numerous studies have found such relationships between social support and emotions [68,69,70].

Specifically, the effects of social support on the health of cancer patients have been examined. In relation to the emotional aspect already mentioned, Wesley et al. [4] found that social support is associated with higher levels of positive emotions and lower levels of negative emotions in an oncology population. Likewise, greater social support has been associated with lower mortality [71], better perceived general health [72] or higher quality of life [3] in people with cancer. In fact, a particular type of social support, such as religious support, has been linked to better mental health in this population [23,73]. However, there is a paucity of research examining social support in relation to trust in God. One study, by Maselko et al. [74], found an association between trust in God and social capital, the latter being understood as one of the social determinants of health, including social support.

The relationship between religion, spirituality and health, though, has been extensively studied. Specifically, numerous studies have focused on cancer and the positive effects of religion and spirituality on the health of these individuals [75,76,77]. In contrast, there are few studies that have examined the influence of trust/mistrust in God on people’s physical and mental health and, to an even lesser extent, the influence of these variables on the health of people with cancer. Thus, in view of the above, it seems that it is necessary to establish clearer relationships between trust/mistrust in God and the emotional and social aspects of cancer patients, due to the implications this may have for oncological patients’ health.

Therefore, the main objective of this study is to test the existence of a relationship between trust/mistrust in God, social support and emotions in oncology patients. In this regard, we propose the following four hypotheses:

**Hypothesis** **1** **(H1).**
*Trust in God will correlate positively with positive emotions, and it will correlate negatively with negative emotions.*


**Hypothesis** **2** **(H2).**
*Mistrust in God will correlate positively with negative emotions, and it will correlate negatively with positive emotions.*


**Hypothesis** **3** **(H3).**
*Social support will correlate positively with positive emotions and negatively with negative emotions.*


**Hypothesis** **4** **(H4).**
*Trust in God, mistrust in God and social support are predictors of positive and negative emotions.*


## 2. Materials and Methods

### 2.1. Study Design and Procedure

A non-probability snowball sampling was used to select the sample, whereby individuals with cancer helped to contact people they knew who also had cancer, although the initial focus of the study was on several cancer patient associations in the city of Salamanca (Spain). The reason for using this type of sampling is due to the potential difficulty in accessing this population, since the associations or organizations that work with them in Spain maintain very strict measures to prevent patients’ disturbance. In addition, protective measures resulting from the COVID-19 pandemic prevented direct contact with participants.

Initially, the participants were sent a questionnaire, which was administered online through the Google Forms platform. The first page of the questionnaire consisted of pertinent information about the characteristics of the study and the questionnaire the participants were asked to answer. The participants were also informed of the confidentiality and anonymity of their responses, as well as the option to stop answering the questionnaire if they felt uncomfortable. Once they received this information and, if they wished to participate in the study, they were offered the option of providing their consent for the confidential treatment of their data for research purposes. After agreeing to participate, they were asked a series of sociodemographic questions. Finally, they had access to the questionnaire. After submitting their responses, each participant helped us to contact other patients, so that we could also distribute the questionnaire to obtain a large sample. Finally, responses from participants who did not meet the inclusion criteria were eliminated. Thus, a total of 28 cases were eliminated from the initial 205 subjects. The ethics committee was blinded for revision purposes.

### 2.2. Sample

A total of 177 people diagnosed with cancer formed the study sample. The participants were 88 men and 89 women, ranging in age from 18 to 81 years (M = 45.7; SD = 16.4). Table 1 shows the sociodemographic characteristics of the sample in more detail.

The criteria for inclusion in the sample were:Cancer diagnosis at the time of study participation.Age greater than 18 years.Accept participation through informed consent.

### 2.3. Measures

#### 2.3.1. Sociodemographic Characteristics and Health Problems

A survey was administered to collect information on various sociodemographic variables: age, sex, educational level, perceived socioeconomic level, employment status and health problems (type of cancer).

#### 2.3.2. Trust/Mistrust in God

The original Brief Trust/Mistrust in God Scale [29] attempts to measure both variables through two subscales of three items each: one for trust in God and one for mistrust in God. The original instrument has a Cronbach’s alpha of 0.94 for the Trust in God subscale and 0.88 for the Mistrust in God subscale. In this study we used the Spanish [78] version, which, like the original, is a Likert-type scale with five response options (from “not at all” to “very much”), in which participants indicate their degree of belief towards the items. It also has three items for the trust in God factor (e.g., “God cares about my deepest concerns”) and three items for the mistrust in God factor (e.g., “God doesn’t care about me”). The higher the score, the greater the degree of trust or mistrust in God, depending on the subscale. This Spanish adaptation has a Cronbach’s alpha of 0.95 for the Trust in God subscale and 0.86 for the Mistrust in God subscale.

#### 2.3.3. Positive and Negative Affect

The Positive and Negative Affect Schedule (PANAS) [79] has been widely used to assess affect and emotions. In fact, it has been previously used in studies to analyze the relationship between trust in God and emotions [46]. This instrument has two subscales: one for positive emotions and the other for negative emotions, with ten items each. In the original version, the former has a Cronbach’s alpha ranging between 0.86 and 0.90, while the latter ranges between 0.84 and 0.87 [79]. In our study, we used the Spanish adaptation of the instrument [80], which is equally composed of ten items for the positive emotions subscale (e.g., “Enthusiastic” or “Interested”) and ten items for the negative emotions subscale (e.g., “Scared” or “Irritable”). It is a Likert-type scale with five response options (from “not at all or very slightly” to “very much”), in which participants must indicate whether they have felt each of the emotions reflected in the items. The score on each subscale ranges from 10 to 50, with higher scores indicating a greater presence of one type of emotion. In this version, Cronbach’s alphas for the positive and negative emotions subscales are 0.92 and 0.88, respectively. In the sample of the present study, the reliability was equally high (α = 0.88 and α = 0.86, respectively).

#### 2.3.4. Perceived Social Support

The Multidimensional Scale of Perceived Social Support, which was originally developed by Zimet et al. [81], was used to assess the level of social support perceived by individuals. This scale has been adapted to Spanish by Landeta and Calvete [82] and Ruiz et al. [83]. As in the case of the original, it consists of twelve items divided into three different dimensions: family, friends and significant others. In fact, only the items of the “significant others” dimension were used in this study, as it represents an even more abbreviated version of measuring perceived social support without focusing on any specific source of support (e.g., “There is one person who is a real source of well-being for me”). The scale is presented in Likert format with seven response options, from “Strongly disagree” to “Strongly agree”. This subscale presents very positive reliability data, with a Cronbach’s alpha of 0.94. In the case of the sample of this study, the reliability data were also adequate (α = 0.80).

### 2.4. Data Analysis

First, the reliability of the instruments was analyzed, obtaining the Cronbach’s alpha of the instruments in the present study. Next, a correlation analysis was performed through Pearson’s r coefficient, in order to examine the relationships between trust/mistrust in God and social support with positive/negative emotions.

Finally, hierarchical regression analyses (forward method) were performed to compare three predictive models, step by step, for each of the dependent variables (PANAS positive and negative emotions). The first model (Model 1) included sociodemographic variables; Model 2 added social support; and, finally, Model 3 added trust/mistrust in God. Given that perceived socioeconomic level, education level and employment status were categorical variables, we created three sets of dummy variables, one for each categorical variable (each set containing k-1 dummy variables for the k levels of the original variables), and then we introduced them as predictors in the regression analyses with positive and negative emotions as dependent variables. The variance inflation factor (VIF) was used as a measure of collinearity, with values greater than 10 being considered problematic [84,85]. SPSS 25 [86] was used for data analysis. Significance level was set at 0.05 for all analyses.

## 3. Results

### 3.1. Relationship between Emotions and Trust/Mistrust in God and Social Support

Positive emotions correlate significantly and positively with trust in God (r = 0.608, *p* < 0.001) and social support (r = 0.474, *p* < 0.001), while negatively with mistrust in God (r = −0.698, *p* < 0.001).

The opposite occurs with negative emotions, which correlate significantly and positively only with mistrust in God (r = 0.660, *p* < 0.001). In contrast, these emotions maintain a significant negative correlation with trust in God (r = −0.617, *p* < 0.001) and social support (r = −0.423, *p* < 0.001).

All of these correlations may be viewed in more detail in Table 2.

### 3.2. Predictive Models of Positive and Negative Emotions

Considering positive emotions as a dependent variable (Table 3), we observed that in Model 1, which includes only the sociodemographic variables, only the lower-middle perceived socioeconomic level (compared to low level) had some explanatory power (for a total of 4.3% of variance explained). For Model 2, in which we added social support to the previous one, we observed that only this variable also explained part of the positive emotions (raising the explained variance up to 23.7%); socioeconomic level was no longer a significant predictor. Finally, when the variables trust/mistrust in God were included in Model 3, we observed that mistrust in God was a significant predictor, and together with social support explained 51.3% of the variance of positive emotions. The more social support the more positive emotions, while mistrust in God scores were negatively related to positive emotions scores.

When we selected negative emotions as the dependent variable (Table 4), we observed that, in Model 1, which only includes the sociodemographic variables, age, perceived socioeconomic level, employment status and educational level explained 21.2% of the variance. Model 2, which added social support to the above, accounted for 32.2% of the negative emotions. Finally, in Model 3, which also includes trust/mistrust in God, we could see that both trust and mistrust in God, together with the above variables (except age) explained 50.7% of the variance of the dependent variable.

In this final model, lower negative emotions scores were related with higher age, the change from low to lower-middle socioeconomic level, higher social support scores and higher trust in God scores.

In addition, higher negative emotions scores were related with the change from unemployed status to pensioner status, the change from no studies to high school (or vocational training) educational level and higher mistrust in God scores.

In any case, the results partially support our hypotheses. It can be observed that trust in God, mistrust in God and social support correlate with positive and negative emotions. In fact, all three variables have predictive power on the negative emotions variable. However, only mistrust in God together with social support were shown to have predictive power on the positive emotions variable.

## 4. Discussion

The aim of this study was to explore the relationships between trust/mistrust in God and social support with positive and negative emotions in cancer patients.

The results of the correlation analyses, first of all, support the first three hypotheses. It can be observed that, indeed, trust in God is positively associated with positive emotions and negatively associated with negative emotions. The opposite happens with mistrust in God, since it maintains a negative relationship with positive emotions and a positive relationship with negative emotions. In other words, religious people, who have a positive and trusting relationship with God, have greater emotional well-being.

These findings are very similar to those obtained in Fadardi and Azadi’s [46] research and thus support the relationship between trust/mistrust in God with indicators of mental health, in this case, positive and negative emotions. In this sense, this study appears to demonstrate similar results as other important studies in this field [30,38,39,40], which show the positive and negative effects of trust and mistrust in God, respectively, on mental health. However, most of these studies focus on the mental health of physically healthy people. Furthermore, our findings shed light on a mental health index, such as positive and negative emotions, in the case of people with cancer, a disease that, from the moment of its diagnosis, has important consequences on psychological health.

Likewise, direct relationships were found between social support and positive emotions, as well as inverse relationships between social support and negative emotions. These results are in line with others that have previously explored these relationships [67,68]. In any case, it seems that social support helps individuals to regulate the emotional impact of complicated circumstances, such as the cancer disease process.

Numerous authors have focused their interest on a specific type of social support, such as religious social support, which refers to social support provided by clergy or other members of a religious organization that is accessed through participation in religious activities [73]. This religious social support is related, for example, to greater well-being [23] or a lower level of depressive symptoms [73], with this type of social support even recognized as predictive of certain aspects of health that general social support does not predict [63]. Therefore, promoting the specific forms of social support that religion and spirituality provide to cancer patients in healthcare settings may have a positive impact on their health.

The findings regarding hypothesis 4, however, invite us to be cautious. When religious variables are added to the models, only mistrust in God proves to have explanatory power over positive and negative emotions. In the case of positive emotions, given the negative nature of the association between both variables, we can state that, in the sample of the present study, lower levels of mistrust in God predict a higher level of positive emotions. The opposite occurs with negative emotions since, given the positive nature of the association between negative emotions and mistrust in God, higher levels of this variable predict higher levels of negative emotions. Moreover, in the case of this type of emotions, it can be observed that trust in God also has explanatory power over these emotions. Thus, given the inverse nature of the relationship between the two variables, we can determine that higher levels of trust in God predict lower levels of negative emotions. Of course, in addition to this, social support helps in predicting positive and negative emotions in oncology patients, improving to some extent the variance explained by trust/mistrust in God.

It is thus observed in our study that, as explained by Rosmarin et al. [31], the relationship between mistrust in God and measures of mental health (such as positive and negative emotions) are stronger than those of trust in God and, for this reason, mistrust in God may have more severe consequences than lack of trust in God. That is, mistrust in God, by implying a negative and even conflicted relationship with God, can have a stronger impact on the emotional well-being of individual than the mere absence of a positive and trusting relationship with God. These consequences probably derive from the relationship proposed by Rosmarin et al. [31] between mistrust in God and divine spiritual struggles. As an aside, inverse relationships have also been found between trust in God and spiritual struggles with God [32]. Be that as it may, spiritual struggles are associated with severe anxiety and depression [87], psychological distress [88] and less positive mental health [89], which could explain, in turn, the serious consequences of mistrust in God. Even in the specific case of cancer, the religious/spiritual struggle was associated with greater symptom burden and poorer quality of life [90], as well as greater depression [91]. At this point, it is worth noting that there are other religious/spiritual constructs that, as in the case of spiritual struggles, have a negative impact on people’s psychological well-being, such as spiritual bypass [13] or guilt [92]. In any case, more research is needed on the serious consequences that mistrust in God may have on people’s health.

Thus, we believe that our results have important potential for the treatment of mistrust in God experienced by patients in health contexts, especially in oncology. That is, the findings bring to light the relevance that the implementation of actions within healthcare settings focused on reducing or alleviating mistrust in God can have, which can in turn reduce the impact it has on the health and well-being of these patients. This is especially relevant given the strong impact that this mistrust seems to have on their emotions and how this may in turn affect their health, not only mentally, but also physically. In this regard, numerous studies have demonstrated the need to take into account various religious and/or spiritual aspects to provide holistic care in cancer care settings [93,94,95,96].

In a related paper consistent with the above discussion, Koenig et al. [20] and Park [97] propose that the relationships between religion, spirituality and health are strongly influenced or mediated by a series of psychological, social and behavioral variables, including positive and negative emotions and social support. We believe that studying the Koenig et al.’s [20] models in depth can help us to better understand the networks that interconnect the variables in this study and others. This, in turn, may assist research in moving toward an improved understanding of the manner in which religion and spirituality influence people’s health.

As an aside, it is important to mention that sociodemographic variables, such as age, perceived socioeconomic level, employment status or educational level, improve the variance explained by the mistrust in God of negative emotions. In general, it has been observed that there are significant relationships between these sociodemographic variables and religiosity and/or spirituality [49]. In addition, studies such as Purborini et al. [51] have found associations between these types of variables and emotions, both negative and positive.

In our study, age had an inverse association with negative emotions, similar to the findings presented by Agrawal et al. [50]. Thus, it appears that young people may present more negative emotions than older people, which may be obvious given the impact of a young person’s life being threatened by a serious illness, such as cancer. Likewise, it may be observed that age is associated with a higher level of trust and a lower level of mistrust in God, i.e., it seems that young people trust God less than older people. This is related to the research of Quinceno and Vinaccia [49], since it is observed that the older the age, the more religious/spirituality people present. Regarding the perceived socioeconomic level, this does not seem to be related to positive emotions. In contrast, lower levels of negative emotions are observed to be related to the change in perceived socioeconomic level from low to lower-middle. In this sense, we found that the higher the perceived socioeconomic level, the lower the level of negative emotions experienced, a relationship that responds to common sense. This idea is consistent with that of Purborini et al. [51], who observed that the higher the socioeconomic level, the less negative emotions appear. In turn, the findings show that higher levels of negative emotions are related to the change in educational level from no studies to high school (or vocational training). That is, it appears that as educational level increases, more negative emotions are experienced. This clashes with what is proposed by authors such as Chiang et al. [52], who have observed the opposite, i.e., a higher educational level is associated with less negative emotions. In addition, the findings show that both perceived socioeconomic level and educational level are associated with a greater mistrust in God. This suggests that people who define themselves as poorer and with a lower level of education are less mistrustful of God. The latter also follows the line of research such as that of Quinceno and Vinaccia [49], who have shown that people with lower socioeconomic and educational levels have better results in religiosity and spirituality. Finally, it was also observed that higher scores of negative emotions are related to a change in employment status from unemployed to pensioner/retired. In this regard, in a systematic review of the literature on employment status and mental health, Hergenrather et al. [98] observed that unemployed people have poorer mental health, although it is appreciated that this improves in the case of retired people. However, this improvement in the mental health of retirees is mostly observed in comparison with employed people. Hergenrather et al. [98] suggest that this difference in the mental health of retirees may vary as a function of the job they held in the past, so research in this respect may be useful. Other research has found that employed people have a higher level of positive emotions, but found no relationship between employment status and negative emotions [99].

Moreover, our data are mostly in agreement with those of other studies that include the variable trust/mistrust in God and that have observed that sociodemographic characteristics significantly predict aspects of mental health such as happiness, anxiety or depression [31]. Therefore, it may be important to consider aspects such as age, educational level, employment status or socioeconomic level when implementing actions related to the treatment of trust/mistrust in God in clinical practice.

Despite the important implications of the findings, it should be mentioned that the study had limitations. First, despite having a large sample, there is a definite difficulty in accessing representative samples when working with oncology patients. With the COVID-19 crisis, accessing these individuals through healthcare settings is complicated and, in addition, turning to associations and organizations that work with this population is very difficult for data protection reasons. In this sense, we believe that these limitations should be considered in future research, as well as the need to study the variables trust/mistrust in God in people affected by other types of pathologies, both physical and mental.

Likewise, studying the religiosity and/or spirituality of individuals implies the need to be particularly careful with the research techniques employed [100]. For this reason, in order to avoid being intrusive in such a subjective and personal aspect of individuals’ lives, we could not utilize numerous measures related to religiosity/spirituality so as to avoid making participants uncomfortable. Nevertheless, the instruments used in this study allowed us to obtain significant data to achieve our aim. Thus, with a view to future research, a qualitative methodology may be useful in this type of work.

An additional consideration is that working with a Spanish oncology population limits us in generalizing our results culturally. However, while this may be one of the limitations of the study, it is also one of its strengths. As stated at the beginning of the study, trust and mistrust in God are variables that have been studied purely in relation to Jewish religiosity, so Rosmarin et al. [31] propose the need to analyze these variables in the Christian population. Thus, this study is a first step in the study of trust in God in the cultural context of a deep Catholic tradition. In any case, further research is needed on these variables with populations of religious faiths in addition to Judaism.

## 5. Conclusions

Through this cross-sectional study, we analyzed the impact that the individual’s relationship with God, based on trust or mistrust, and social support have on the emotional health of people affected by cancer, a disease with important physical and psychological consequences. Broadly speaking, it was observed that trust in God and social support have a positive impact on the affective well-being of cancer patients, in contrast to mistrust in God. In any case, this research sheds light on the role of trust/mistrust in God and social support on the emotions of the individuals, which provides clinical relevance to our findings. In this sense, we believe that it is necessary to address religiosity and spirituality in health contexts, since it constitutes a psychosocial and cultural variable that may have particularly favorable repercussions on health.

## Figures and Tables

**Table 1 healthcare-10-01138-t001:** Sociodemographic characteristics. *N* = 177.

Variables		*N*	%
Age ^1^		45.7	16.4
Gender	Man	88	49.7
	Woman	89	50.3
Education level	No studies	5	2.8
	Elementary education	10	5.6
	Secondary education	10	5.6
	Vocational training or high school	42	23.7
	Higher education	70	39.5
	Postgraduate, master’s or doctoral degree	40	22.6
Employment status	Currently working	74	41.8
	On the dole	12	6.8
	Not working	38	21.5
	Student	28	15.8
	Pensioner/Retired	25	14.1
Perceived socioeconomic level	Low	1	0.6
	Lower-middle	44	24.9
	Average	86	48.6
	Upper-middle	44	24.9
	High	2	1.1
Main types of cancer	Breast cancer	36	20.3
	Leukemias ^2^	29	16.4
	Colon cancer	17	9.6
	Lung cancer	15	8.5
	Hodgkin’s lymphoma	11	6.2
	Non-Hodgkin’s lymphoma	6	3.4

^1^ Mean and Standard Deviation (SD). ^2^ Leukemias include lymphoblastic leukemia, myeloblastic leukemia and unspecified leukemias.

**Table 2 healthcare-10-01138-t002:** Correlations of PANAS positive and negative emotions.

Variables	1	2	3	4	5
1. Positive emotions		−0.684	0.608	−0.698	0.474
2. Negative emotions	−0.684		−0.617	0.660	−0.423
3. Trust in God	0.608	−0.617		−0.828	0.442
4. Mistrust in God	−0.698	0.660	−0.828		−0.479
5. Social support	0.474	−0.423	0.442	−0.479	

Note: All correlations are significant at the *p* < 0.001 level.

**Table 3 healthcare-10-01138-t003:** Predictive models of PANAS positive emotions.

							95% CI		
Model	R^2^	Predictor	B	SE	t	*p*	LL	UL	FIV
1	0.043	Intercept	32.233	0.533	60.515	0.000	31.182	33.284	
		Lower-middle perceived socioeconomic level	2.994	1.068	2.803	0.006	0.886	5.103	1.000
2	0.237	Intercept	−1.464	5.081	−0.288	0.774	−11.492	8.563	
		Lower-middle perceived socioeconomic level	1.659	0.977	1.698	0.091	−0.269	3.588	1.044
		Social support	1.295	0.194	6.662	0.000	0.912	1.679	1.044
3	0.513	Intercept	26.207	4.938	5.307	0.000	16.460	35.953	
		Lower-middle perceived socioeconomic level	0.249	0.796	0.313	0.754	−1.321	1.820	1.078
		Social support	0.516	0.175	2.957	0.004	0.172	0.861	1.310
		Mistrust in God	−0.963	0.097	−9.900	0.000	−1.155	−0.771	1.340

Note: Only significant models are shown; Model 1 included sociodemographic variables; Model 2 included the previous variables and social support; Model 3 included all the previous variables and trust/mistrust in God. The category of reference for perceived socioeconomic level was ‘low’.

**Table 4 healthcare-10-01138-t004:** Predictive models of PANAS negative emotions.

								95% CI		
Model	R^2^	Predictor	B	SE	Beta	t	*p*	LL	UL	FIV
1	0.212	Intercept	24.650	1.421		17.341	0.000	21.844	27.455	
		Age	−0.135	0.031	−0.371	−4.302	0.000	−0.197	−0.073	1.662
		Lower-middle perceived socioeconomic level	−4.184	0.959	−0.303	−4.363	0.000	−6.077	−2.291	1.078
		Employment: Pensioner/Retired	3.467	1.460	0.202	2.375	0.019	0.586	6.349	1.622
		Education level: High school	2.241	0.977	0.160	2.292	0.023	0.311	4.170	1.085
2	0.322	Intercept	48.964	4.710		10.397	0.000	39.668	58.261	
		Age	−0.123	0.029	−0.338	−4.215	0.000	−0.181	−0.066	1.672
		Lower-middle perceived socioeconomic level	−3.389	0.902	−0.245	−3.759	0.000	−5.169	−1.610	1.107
		Employment: Pensioner/Retired	4.227	1.361	0.247	3.105	0.002	1.540	6.914	1.639
		Education level: High school	2.148	0.907	0.153	2.369	0.019	0.358	3.938	1.085
		Social support	−0.957	0.178	−0.348	−5.378	0.000	−1.308	−0.606	1.090
3	0.507	Intercept	28.732	5.156		5.573	0.000	18.554	38.909	
		Age	−0.055	0.026	−0.150	−2.077	0.039	−0.107	−0.003	1.864
		Lower-middle perceived socioeconomic level	−2.355	0.780	−0.171	−3.018	0.003	−3.895	−0.815	1.140
		Employment: Pensioner/Retired	3.907	1.175	0.228	3.325	0.001	1.588	6.227	1.679
		Education level: High school	1.792	0.778	0.128	2.304	0.022	0.257	3.327	1.098
		Social support	−0.374	0.168	−0.136	−2.227	0.027	−0.706	−0.042	1.336
		Mistrust in God	0.535	0.149	0.354	3.586	0.000	0.241	0.830	3.473
		Trust in God	−0.292	0.134	−0.214	−2.169	0.031	−0.557	−0.026	3.487

Note: Forward method. Only significant variables included in the models are shown; Model 1 included sociodemographic variables; Model 2 included the previous variables and social support; Model 3 included all the previous variables and trust/mistrust in God. The category of reference for perceived socioeconomic level was ‘low’. The category of reference for employment status was ‘unemployed’. The category of reference for the education level was ‘no studies’.

## Data Availability

The data presented in this study are available upon request from the corresponding author.

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
