# Peer review of "Religiosity, Emotions and Health: The Role of Trust/Mistrust in God in People Affected by Cancer"

_healthcare, 2022, doi:10.3390/healthcare10061138_

Round 1
Reviewer 1 Report
Dear authors. Congratulations, for the work well done. Good for you and good of you. It is truism that God implies the conviction that God looks after a person's own interests (line 14). I fully agree with you.
Additional comments
1. It is a fact that Religion and Spirituality has sparked debate across the globe.
2. The title is short, precise and to the point; and is line with the research objectives of the study.
3. Good referencing is commendable, however, Bibliography/List of references needs attention (alphabetical order), unless the layout/structure is recommended by the Journal.
4. Conclusion is too short. The authors could consider adding at least six more sentences.
5. Generally, well researched topic & well focused and organized discussion supported by relevant and sufficient sources.
6. This study will add value in a scholarly arena.
Reviewer 2 Report
The article deals with an undoubtedly far-reaching and inspirational topic, namely the potential of religiosity in cancer treatment. I would suggest the article for publication even as it is, though some more minute specifications could further improve the quality.
In my opinion, the abstract should mention that the target group of examined patients came from Spain – which is otherwise an excellent choice, a traditionally Catholic nation indeed, as the article correctly observes. Similarly, I believe the abstract should stress that this is the first research of this kind in a Christian context. The comparative religious aspects of the article are among its potentially most influential results, pointing out that religiosity can improve physical and mental health independently of the type of religion.
Very importantly, though: is the aim of the article the treatment of trust/mistrust in God as claimed in ll. 408-9 and 468; or the treatment of cancer…? In other words, are the motivations behind this research inspired by religious feelings or by scientific ambitions…?
From the otherwise very thorough definition of the basic concepts used in the article, I miss the clarification of the difference between ‘religiosity’ and ‘trust in God.’ Similarly, please clarify the difference between ‘mistrust’ and ‘lack of trust’ – does either equal ‘distrust’? If I understand the authors’ concepts well, it may not be fully irrelevant that mistrust in God is already faith, meaning a belief that God exists but is not necessarily willing to help the individual in a given situation, in this case cancer.
In lack of a better fitting category, I rated the originality of the article as 'average' because at times I had the feeling that the article stated the intuitively obvious at the comparison of certain social groups. Of course younger people experience cancer more negatively than older people: they are about to lose an entire life. Of course economically more stable groups are less worried about cancer: they have more financial means to access treatments. It is definitively an advantage, though, if an article can support intuitive opinions with proper scientific research results.
Having said that, I still have some doubts regarding the validity of the ‘retired’ vs. ‘unemployed’ comparison. The former group have a solid income, whereas the latter do not, which is a defining difference in their socioeconomic status and consequent access to medical treatments.
I believe it would help the clear presentation of the results if the findings were summarized in just very simple words, e.g. ‘religious people cope better with the emotional burden of cancer’.
From the point of view of language, the consistent use of ‘on the other hand’ as a linker of contrast (ll. 72, 98, 131, 150, 333, 362, 365, 442) is somewhat disturbing. It is not strictly speaking a mistake but to the best of my understanding, top quality English essays tend to avoid using ‘on the other hand’ without ‘on the one hand’.
Overall, I still find this a very engaging article with great potential on a topic of wide public interest, and definitely recommend it for publication. Thank you for the opportunity of peer-reviewing it, please do not hesitate to contact me for any further clarification.
Reviewer 3 Report
Thank you for the opportunity to review this paper. Overall, it is researched and written with rigour, expertise and a strong discovery and understanding of previous allied research.
I am not qualified to assess your statistical analysis, nevertheless I have some concerns around your characterizations of religion and spirituality that you may address if you find them relevant. 1) I found the final discussion surprisingly discursive and inconclusive, as you speculated on the significance of combinations of variables. This may be because of the unexpected finding related to Hypothesis 4, as discussed in lines 392-407, but I wondered if it was also to do with your choice, definition and discussion of key terms as part of the research design.
For instance 2) Why was trust/mistrust in God the basic binary for your research? Both assume the existence of a God, rather than allowing that one might be agnostic, or not believe in a personal God, and yet still be religious. If only the trust/mistrust pairing is used to develop Likert scale questions it seems intuitively likely that to believe God exists and yet is 'wilfully ignorant and malevolent' [of or towards one's wellbeing] would be productive of negative emotions and their consequences. (Lines 83-85)
3) To define spirituality (only) as 'a characteristic of deeply religious people' is reductive. Woodhead & Heelas's 2000, Religion in Modern Times: an Interpretative Anthology for instance, outlines a range of contemporary models of religiosity from traditional Religions of Difference (through Religions of Humanity (most of the world religions occur in both those forms) to Spiritualities of Life which come in both expressive and utilitarian forms. All along that range, whether the manifestation is theistic or non-theistic in nature, there is the possibility of sufficient 'spirituality' being accessible to produce positive (or at least non-negative) emotions in medically adverse circumstances. Anna King's 'Spirituality: Transformation and Metamorphosis' (1996) has also provided an influential assessment of the manifold meanings and applications of the term 'spirituality'. I realize the researchers were working within a specific culture still strongly marked by a Religion of Difference, but it is not clear whether all the participants were actually committed to that religion and to a specific concept of God and spirituality.
Around line 159 I also wanted a better discussion of what 'social support' might entail - the assumption seemed to be that reader would know what the options were. 'Religious support' was identified and dismissed as relevant in this case, which seems odd, while in lines 269-272 it was stated that a set of measures of social support was 'abbreviated' to just one dimension relating to significant others. Since discussion of social support as a factor interactive with other variables came up again several times in the concluding discussion, I wondered if its abbreviation to a single dimension had foreclosed on the possibility of more informative findings.
215-218 If participants could send the questionnaire on to other participants how was confidentiality maintained during that process? Maybe just a matter of explaining how that worked.
